# Collateral Circulation and BNP in Predicting Outcome of Acute Ischemic Stroke Patients with Atherosclerotic versus Cardioembolic Cerebral Large-Vessel Occlusion Who Underwent Endovascular Treatment

**DOI:** 10.3390/brainsci13040539

**Published:** 2023-03-24

**Authors:** Ruoyao Cao, Yao Lu, Peng Qi, Yanyan Wang, Hailong Hu, Yun Jiang, Min Chen, Juan Chen

**Affiliations:** 1Department of Radiology, Beijing Hospital, National Center of Gerontology, Institute of Geriatric Medicine, Chinese Academy of Medical Sciences, Beijing 100730, Chinacjr.chenmin@vip.163.com (M.C.); 2Graduate School of Peking Union Medical College, Beijing 100730, China; 3Department of Neurosurgery, Beijing Hospital, National Center of Gerontology, Institute of Geriatric Medicine, Chinese Academy of Medical Sciences, Beijing 100730, China; 4Department of Neurology, Beijing Hospital, National Center of Gerontology, Institute of Geriatric Medicine, Chinese Academy of Medical Sciences, Beijing 100730, China

**Keywords:** collateral circulation, B-type natriuretic peptide, acute ischemic stroke, endovascular treatment, clinical outcome

## Abstract

Purpose: The aim of this study was to verify the value of collateral circulation and B-type natriuretic peptide (BNP) in predicting clinical outcomes of patients with acute ischemic stroke (AIS) and their biomarker value for stroke subtypes before endovascular treatment (EVT). Patients and Methods: In this retrospective study, 182 patients who underwent EVT for unilateral anterior circulation large-vessel occlusion between March 2016 and January 2022 were analyzed. The modified collateral circulation scoring system on four-dimensional computed tomography angiography (4D CTA-CS) was used to assess collateral status, and stroke subtypes were determined according to the TOAST classification criteria. Patients were divided into good (mRS ≤ 2) and poor outcome (mRS > 2) groups based on their modified Rankin Scale (mRS) score at 3 months. Results: 4D CTA-CS was an independent predictor of the clinical outcome for all AIS patients (odds ratio = 0.253; 95% CI, 0.147–0.437; *p* < 0.001), CE stroke patients (odds ratio = 0.513; 95% CI, 0.280–0.939; *p* = 0.030), and LAA stroke patients (odds ratio = 0.148; 95% CI, 0.049–0.447; *p* = 0.001). The BNP was a biomarker for clinical outcome prediction in CE (odds ratio = 1.004; 95% CI, 1.001–1.008; *p* = 0.005) but not in LAA patients. Combined with BNP, 4D CTA-CS improved predictive values for clinical outcomes (*p* < 0.05). Conclusion: Collateral status and BNP could be used as independent predictors of clinical outcomes in AIS patients and could determine stroke subtypes (CE stroke or LAA stroke). In addition, the model of 4D CTA-CS combined with BNP was the most effective in predicting clinical outcomes compared with collateral status or BNP alone.

## 1. Background

The management of acute large-vessel occlusion has made significant breakthroughs with the advancement of endovascular treatment (EVT) [1]. Acute ischemic stroke (AIS) has multiple etiological subtypes, among which large-artery atherosclerosis (LAA) and cardiogenic embolism (CE) are the two primary causes of intracranial large-vessel occlusion [2]. Although the clinical features observed at stroke onset can be helpful in distinguishing the two subtypes of stroke (LAA and CE subtype), it is still unclear how imaging and laboratory information can contribute to the prognosis assessment of AIS and its subtypes, particularly in the evaluation of the prognosis after EVT [3,4]. Therefore, it is necessary to clarify the prognostic risk factors of different AIS subtypes so as to provide a reference for optimizing EVT of AIS.

Several randomized clinical controlled trials have confirmed that the identification of suitable AIS patients for EVT using imaging evaluation is crucial to obtaining a favorable clinical outcome [5]. Particularly, multimode CT, including four-dimensional CT angiography and CT perfusion (4D CTA-CTP), can provide a comprehensive morphological and functional assessment of collateral circulation in AIS [6]. Collateral circulation is not only the main factor determining the final infarct volume and ischemic penumbra of AIS, but also the major factor affecting the final functional outcome of patients with EVT [7,8]. However, most previous studies included patients with different stroke subtypes in one group when exploring the effect of collateral status on the clinical outcome. Emerging evidence has indicated the pathogenesis of LAA stroke is different from that of CE stroke, which may have different effects on the formation and characteristics of collateral circulation, as well as on clinical outcome [9,10].

B-type natriuretic peptide (BNP), a vasoactive hormone mainly synthesized by myocardial tissue, is widely used in the prediction and diagnosis of heart failure [11]. The increase in blood BNP level in the early stage of AIS has been reported to be associated with poor functional outcomes of patients receiving intravenous thrombolysis [12]. Nevertheless, there are few studies focused on the correlation between serum BNP level at admission and outcome in AIS patients undergoing EVT.

Herein, we retrospectively analyzed the data of AIS patients undergoing EVT. Four-dimensional CTA-CTP imaging technology was used to evaluate the prognostic ability of collateral status for AIS and to determine stroke subtypes (LAA stroke and CE stroke) before EVT. Meanwhile, the feasibility of cerebral collateral circulation combined with serum BNP in predicting the clinical outcome of AIS patients receiving EVT was explored.

## 2. Patients and Methods

### 2.1. Study Population

We conducted a retrospective analysis of AIS patients who received EVT in our hospital from March 2016 to January 2022. The study was approved by the local institutional research ethics board, and informed consent was waived since the patients’ data were evaluated retrospectively and anonymously.

Inclusion criteria for the study were [13]: (1) age ≥ 18 years old; (2) one-stop 4D CTA-CTP examination revealed large-vessel occlusion of unilateral anterior circulation; (3) EVT performed within 24 h after symptom onset. (According to the guidelines, in patients with anterior circulation large-vessel occlusion presenting within 6–24 h from symptom onset, one-stop CTA-CTP imaging can be used to select patients for endovascular treatment. This includes patients with a small core infarct volume and a large perfusion mismatch ratio.)

Exclusion criteria were [13]: (1) intracranial hemorrhage detected by non-contrast CT scan (NCCT); (2) previous large cerebral infarction of cerebral hemisphere; (3) moderate/severe stenosis and/or occlusion of contralateral arteries; (4) incomplete clinical, laboratory, or imaging data.

### 2.2. Data Collection

#### 2.2.1. Imaging Data Collection

Initially, NCCT was performed on all patients using scan parameters of 135 kV and 300 mAs to rule out intracranial hemorrhage. Subsequently, a one-stop whole-brain dynamic-volume 4D CTA-CTP examination was performed using a 16cm wide-detector CT scanner (Aquilion ONE, Canon Medical Systems, Otawara, Japan). Patients were given a non-ionic iodinated contrast medium (Iopamidol, Braccosine, Shanghai, China) intravenously at a dosage based on their body weight (0.6 mL/kg), followed by 30 mL saline using a two-channel high-pressure injector. The dynamic volume perfusion scan was carried out 7 s after contrast injection with scan parameters of 80 kV and 100 mAs. The coverage area was 160 mm, and the layer thickness was 0.5 mm [13].

The following imaging information was collected: ASPECTS scores [14], ischemic penumbra (IP) volume, ischemic core (IC) volume, the mismatch ratio (MMR) [15], the modified collateral circulation scoring system on 4D CTA (4D CTA-CS) scores [8], clot burden score (CBS) scores [16], and the final infarct volume (FIV) [15].

#### 2.2.2. Clinical Data Collection

The study collected various information of patients who underwent EVT for large-vessel occlusion, including the National Institutes of Health Stroke Scale (NIHSS) score, age, gender, time information, risk factors of cerebrovascular disease (such as hypertension, diabetes mellitus, hyperlipidemia, coronary heart disease, previous stroke, atrial fibrillation, and smoking), laboratory test results (such as blood fibrinogen, INR, serum BNP, etc.), stroke types based on Trial of ORG 10,172 in Acute Stroke Treatment (TOAST) classification [17], and good or poor outcomes based on mRS score (0–2 for good and 3–6 for poor).

#### 2.2.3. Related Factors for EVTs

Decisions of EVTs were made based on one-stop 4D CTA-CTP results, with conscious sedation or local anesthesia administered as needed. EVT strategies included intra-arterial thrombolysis, stent retriever, contact aspiration, a combination of stent retriever and aspiration, and percutaneous transluminal angioplasty and/or stenting. Recanalization was evaluated using the modified Thrombolysis In Cerebral Ischemia criteria (mTICI), with successful recanalization defined as mTICI grade ≥ 2b [18].

### 2.3. Statistic Analysis

Non-normal distribution quantitative data were presented as median (Q1, Q3). Qualitative data were presented as frequency (percentage). Both parametric and non-parametric tests were employed to account for skewed data. Differences in qualitative data between groups with good (mRS ≤ 2) and poor (mRS > 2) outcomes were compared using Chi-square tests. Non-normal quantitative data for differences between the good and the poor outcome group were compared using non-parametric Mann–Whitney U tests. Logistic regression was used to identify risk factors for clinical outcomes, with variable selection based on Akaike Information Criterion (AIC). To evaluate the predictive value of 4D CTA-CS, serum BNP, and their combination for clinical outcomes, receiver operating characteristic (ROC) curves were used. The area under the curve (AUC) of the ROC for 4D CTA-CS, BNP, and their combination was compared using the DeLong test. Statistical significance was defined as *p* < 0.05 (two-tailed). All analyses were performed using Statistical Package for the Social Sciences ver. 25 software (SPSS, Chicago, IL, USA).

## 3. Results

### 3.1. Baseline Characteristics

In total, 182 patients were enrolled in this study, including 103 males (56.60%) and 79 females (43.40%). According to TOAST classification, 77 patients had CE stroke and 105 patients had LAA stroke. Compared with those in the poor outcome group, the patients with good outcomes showed better collateral status and lower serum BNP (all, *p* < 0.05). Detailed information is summarized inTable 1, Table 2, Appendix A.

Figure 1 shows the distribution of the collateral status, serum BNP level, and mRS scores in all the patients, as well as in the CE and LAA stroke subgroups.

### 3.2. The Predictive Values of Baseline Factors in Poor Outcome (mRS > 2) Group and Stroke Subtypes

Table 3 shows the adjusted odds ratios (ORs) for predicting clinical outcome. After adjusting for age and gender, both serum BNP level (OR = 1.003; 95% confidence interval (CI), 1.001–1.005; *p* = 0.001) and 4D CTA-CS (OR = 0.281; 95% CI, 0.170–0.466; *p* < 0.001) were found to be independent predictive factors of clinical outcome for all patients. For CE stroke patients, serum BNP (OR = 1.004; 95% CI, 1.001–1.008; *p* = 0.005) and 4D CTA-CS (OR = 0.513; 95% CI, 0.280–0.939; *p* = 0.030) were independent predictive factors. For LAA stroke patients, 4D CTA-CS (OR = 0.209; 95% CI, 0.090–0.488; *p* < 0.001) was the only independent predictive factor. After further adjusting for risk factors, both serum BNP level (OR = 1.004; 95% CI, 1.002–1.006; *p* = 0.001) and 4D CTA-CS (OR = 0.244; 95% CI, 0.138–0.432; *p* < 0.001) remained independent predictive factors for all patients. For CE stroke patients, serum BNP (OR = 1.007; 95% CI, 1.003–1.012; *p* = 0.003) and 4D CTA-CS (OR = 0.395; 95% CI, 0.179–0.873; *p* = 0.022) were independent predictive factors. For LAA stroke patients, 4D CTA-CS (OR = 0.122; 95% CI, 0.035–0.425; *p* = 0.001) was the only independent predictive factor.

### 3.3. Diagnostic Performance of Serum BNP and 4D CTA-CS

The results are shown in Table 4 and Figure 2: the AUCs of serum BNP in all patients and the CE stroke group were 0.792 [95% CI (0.726–0.849), *p* < 0.001)] and 0.751 [95% CI (0.639–0.842), *p* < 0.001], respectively. The cut-off value of serum BNP level in predicting the clinical outcome of all patients was 195.51 pg/mL, with a sensitivity of 79.52% and a specificity of 68.69%, and that of CE stroke patients was 470.58 pg/mL, with a sensitivity of 62.50% and a specificity of 79.31%. Compared with BNP or 4D CTA-CS alone, the combination of 4D CTA-CS and BNP improved prognostic ability (both *p* < 0.05). The AUCs of this combination were 0.870 (sensitivity 79.52%, specificity 85.86%), 0.863 (sensitivity 83.33%, specificity 82.76%), and 0.870 (sensitivity 74.29%, specificity 87.14%) for all patients, CE stroke patients, and LAA stroke patients, respectively.

All patients were divided into two groups according to the collateral status: the good collateral group (3–4 scores) and the poor collateral group (0–2 scores) (Appendix A). The proportion of good outcomes was much higher in patients with good 4D CTA-CS scores than in those with poor 4D CTA-CS scores for all patients (78.45% vs. 21.55%), CE stroke patients (66.67% vs. 33.33%), and LAA stroke patients (83.75% vs. 16.25%).

In addition, the patients were also divided into four groups according to the cut-off values of serum BNP level. BNP ≥ 195.51 pg/mL in all patients and BNP ≥ 470.58 pg/mL in CE stroke patients suggested a higher proportion of poor outcomes (Appendix A).

## 4. Discussion

This study retrospectively analyzed 182 AIS patients who underwent EVT within 24 h from symptom onset, and the clinical outcome and prognostic factors of all patients, as well as those of patients with CE stroke or LAA stroke were discussed. The results demonstrated that collateral status and serum BNP could be used as independent predictors of AIS and indicators of stroke subtypes. In addition, this study revealed that the model of 4D CTA-CS combined with BNP was the most effective in predicting the clinical outcome of AIS and in prompting stroke subtypes (CE group or LAA group).

Consistent with our previous studies, the present study confirmed that good collateral circulation contributed to the good clinical outcome of patients receiving EVT [15]. Various carotid artery imaging techniques, such as ultrasound, are helpful in identifying vessel and collateral status. Ultrasound is a valuable tool in evaluating patients with severe carotid artery stenosis and recent ischemic stroke [18]. This diagnostic modality complements CTA by providing valuable information on blood flow dynamics. Transcranial Doppler ultrasound, in particular, is a valuable tool for assessing collateral circulation, as it can provide information on blood flow velocities and patterns in the cerebral arteries. This information is essential in determining the adequacy of collateral circulation and guiding clinical management decisions [18,19]. However, ultrasound is less sensitive and specific than CTA in detecting large-vessel occlusions, and the quality of images depends on the operator’s experience and skill [20,21]. Given the critical importance of accurate diagnosis in AIS patients, this study employed a one-stop CTA-CTP examination. In this study, further subgroup analysis showed that patients with good collateral status in either the LAA group or the CE group had a higher proportion of good outcomes than patients with poor collateral status. After adjusting for confounding factors such as gender, age, and risk factors, CTA-CTP indicated that good collateral vessels exerted a protective effect on the clinical outcomes of patients with EVT. Accordingly, better collateral circulation is related to smaller core infarction and a larger mismatch ratio, which allows for the more ischemic brain tissue to remain in the penumbra state but not undergo irreversible changes, providing more opportunities for rescue by EVT [22]. Collateral circulation can clear micro embolism, promote endogenous fibrinolytic substances or exogenous fibrinolytic drugs to reach the embolism, and improve the recanalization rate of EVT [23]. Our results also revealed that the proportion of good collateral circulation in the LAA group was higher than that in the CE group, which was due to the different collateral circulation formation processes between LAA stroke and CE stroke [9]. LAA is the formation of in situ thrombosis related to long-term chronic vascular stenosis or plaque rupture [24]. When severe intracranial vascular stenosis or occlusion leads to anterograde blood flow restriction, the pressure gradient caused by hemodynamic changes triggers vascular inflammatory responses, promotes the activation of the nitric oxide (NO) signal pathway, facilitates vascular reconstruction, and initiates secondary and tertiary collateral circulation [25]. CE originates from the embolus falling from the formed thrombus in the heart under certain conditions; it circulates through the blood to the intracranial artery and results in vascular occlusion. Because of the instantaneous process, secondary and tertiary collateral circulation is not likely to develop immediately [9,26,27]. For a long time, it has been believed that compared with acute vascular occlusion caused by embolization, in situ thrombosis caused by vascular stenosis has a rapid and comprehensive opening of collateral branches due to the basis of chronic stenosis. The speed of vascular occlusion is one of the key factors affecting the compensatory ability of pial collateral circulation. Previous studies have shown that patients with LAA stroke have better cranial branch circulation, less thrombus load, larger salvable brain tissue, and better prognosis than patients with CE stroke [3,9,10,27].

In addition to exploring imaging markers that can predict prognosis, this study sought to find blood markers with prognostic values. The brain stem, cerebral cortex, and subcortical forebrain area constitute a complex neural network with the cardiovascular system. The close relationship between brain tissue and the cardiovascular system is constantly being explored. BNP, a neurohormone with similar structure and function to the atrial natriuretic peptide, is of great significance in the pathogenesis, diagnosis, treatment, and prognosis of cardiovascular and cerebrovascular diseases [28]. At present, the correlation between AIS and plasma BNP has been indicated. Clinical studies have reported that blood BNP is related to the clinical outcomes of AIS, that is, elevated serum BNP is related to increased mortality and poor outcome after stroke [12,29]. However, there are few studies on the correlation between the clinical outcome and the admission BNP level of AIS patients undergoing EVT.

The findings of this study suggest that BNP can serve as a reliable blood marker for prognosticating the clinical outcome of AIS patients. Moreover, the study highlighted a significant variation in BNP levels between the LAA and CE subtypes of AIS. Multivariate analysis showed that BNP in the CE group, but not in the LAA group, had a predictive value in the clinical outcome. Some of the main reasons are as follows: (1) Atrial fibrillation is a high-risk factor leading to acute cerebral infarction, and most emboli in CE patients are caused by persistent or paroxysmal atrial fibrillation. The atrium is the main source of circulating BNP in patients with atrial fibrillation. Atrial stretch and excessive volume load result in increased BNP release [11,29]. (2) Patients with CE stroke have congestive heart failure before the acute onset of stroke. Studies have shown that patients with congestive heart failure have a two- to three-fold increased risk of stroke. Abnormal coagulation mechanism, dysfunction of endothelial cells, and aberrant blood components are also the mechanisms of stroke in patients with heart failure. Therefore, congestive heart failure may further lead to CE stroke. As an important index for patients with congestive heart failure, BNP is elevated in patients with CE stroke [30,31]. (3) When stroke occurs, the nervous system and endocrine system will increase the load of the ventricular wall, further increasing the possibility of congestive heart failure. Therefore, blood BNP level is elevated more apparently in CE stroke patients.

In this study, the ROC curve showed that compared with 4D CTA-CS or BNP alone, the combination of these two indexes optimized the ability to predict the clinical outcome of AIS and could indicate stroke subtypes after EVT. Four-dimensional CTA can more intuitively and comprehensively reflect the collateral circulation state and is simple, practical, and repeatable. Meanwhile, it makes up for the deficiency of conventional single-phase CTA. BNP is highly expressed in stroke patients, especially in CE stroke, with high diagnostic sensitivity and specificity, fast and convenient clinical detection, and affordable cost. A predictive model combining these two indexes can assist clinicians in the early identification of AIS patients with poor outcomes.

This study has several limitations. Firstly, this is a single-center clinical study with a relatively small sample size, which prevented us from conducting further subgroup analysis on collateral compensation differences for other TOAST classification (small-vessel disease and other stroke subtypes), and patients with recent and long-term stroke were not discussed in this study. Additionally, the retrospective nature of the study and lack of a randomized control group may introduce selection bias. Hence, further research using large-scale, prospective, and randomized controlled multi-center clinical studies is necessary to validate our findings. Therefore, a promising area for future research would explore collateral circulation and BNP as predictors of outcomes in diverse subtypes of ischemic stroke [32].

## 5. Conclusions

In summary, collateral status and BNP can be used as independent predictors of clinical outcome in AIS and can be helpful in identifying different stroke subtypes (CE stroke and LAA stroke). The model of 4D CTA-CS combined with BNP is mostly effective in predicting the clinical outcome of AIS when compared with 4D CTA-CS or BNP alone and might determine stroke subtypes (CE or LAA stroke).

## Figures and Tables

**Figure 1 brainsci-13-00539-f001:**
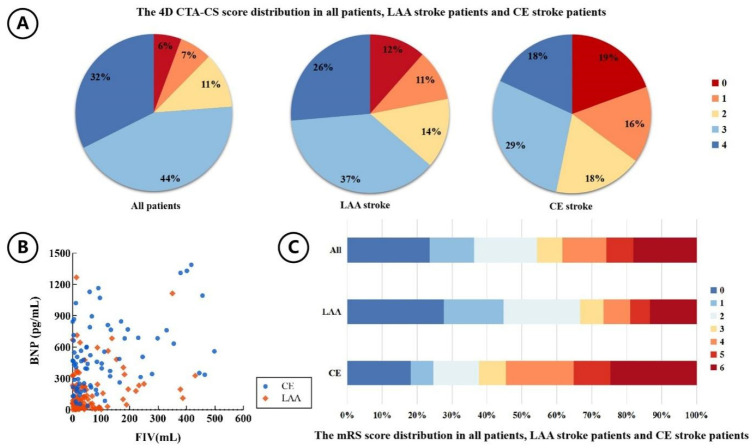
(**A**) The pie charts show the distribution of collateral grading was different in all patients, LAA stroke patients, and CE stroke patients. The proportion of good collateral in LAA stroke patients was much higher than that in CE stroke patients. (**B**) The scatter plot shows the distribution of serum BNP level was different between CE stroke patients and LAA stroke patients. The level of serum BNP in CE stroke patients was much higher than that in LAA stroke patients. (**C**) The bar charts show the distribution of mRS scores in all patients, LAA stroke patients, and CE stroke patients. The proportion of good outcomes in LAA stroke patients was higher than that in CE stroke patients.

**Figure 2 brainsci-13-00539-f002:**
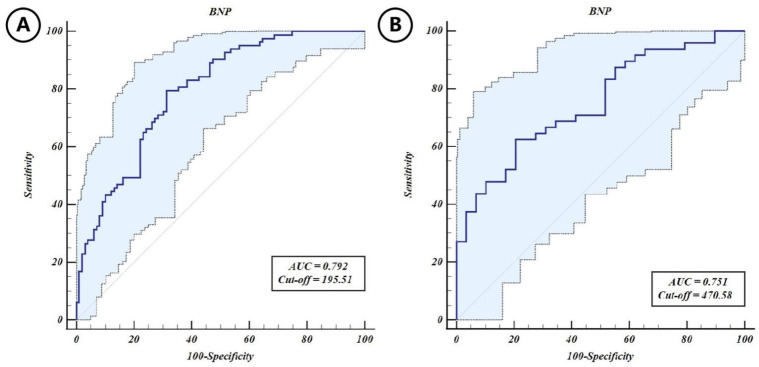
ROC curve analysis for the relationship of mRS scores with serum BNP in all stroke patients and CE subtype patients. (**A**) In all patients, the area under the curve (AUC) of this model was 0.792 (95% confidence interval (CI), 0.726–0.849). The cut-off value of serum BNP level for all patients was 195.51 pg/mL, with sensitivity of 79.52% and specificity of 68.69%. (**B**) In CE stroke patients, this model resulted in an AUC of 0.751 (95% CI, 0.639–0.842). The cut-off value of serum BNP level for CE stroke patients was 470.58 pg/mL, with a sensitivity of 62.50% and specificity of 79.31%.

**Table 1 brainsci-13-00539-t001:** Patient characteristics at baseline.

Characteristic	All Patients (*n* = 182)	Good Outcome (mRS ≤ 2) (*n* = 99)	Poor Outcome (mRS > 2) (*n* = 83)	*p*-Value
Age, y; median (IQR)	76.00 (63.75, 83.25)	70.00 (59.00, 80.00)	81.00 (72.00, 85.00)	<0.001 ***
Female, n (%)	79 (43.40)	34 (34.34)	45 (54.22)	0.010 *
NIHSS, median (IQR)	12.50 (7.00, 17.00)	9.00 (6.00, 15.00)	15.00 (11.00, 19.00)	<0.001 ***
IV-Tpa, n (%)	43 (23.63)	27 (27.27)	16 (19.28)	0.224
SBP, median (IQR)	145.00 (132.75,159.25)	143.00 (130.00,152.00)	146.00 (135.00,146.00)	0.079
DBP, median (IQR)	80.00 (71.00, 90.00)	80.00 (71.00, 88.00)	80.00 (71.00, 94.00)	0.320
Risk factors, n (%)				
Smoking	53 (29.12)	37 (37.37)	16 (19.28)	0.009 **
AF	75 (41.21)	26 (26.27)	49 (59.04)	<0.001 ***
Hypertension	140 (76.92)	74 (74.75)	66 (79.52)	0.484
Diabetes mellitus	73 (40.11)	36 (36.36)	37 (44.58)	0.290
Hyperlipidemia	83 (45.60)	45 (45.45)	38 (45.78)	1.000
CHD	79 (43.41)	33 (33.33)	46 (55.42)	0.004 **
Previous stroke	82 (45.05)	39 (39.39)	43 (51.81)	0.102
Imaging examination, median (IQR)/n (%)				
IC volume, mL	22.55 (8.99, 57.65)	19.28 (7.48, 37.10)	45.82 (14.82, 113.56)	<0.001 ***
IP volume, mL	82.94 (42.92, 128.05)	81.74 (33.05,128.23)	87.09 (50.33,127.99)	0.407
MMR	3.04 (1.76, 6.25)	3.54 (2.23, 7.20)	2.21 (1.07, 4.00)	<0.001 ***
FIV, mL	39.44 (11.05,101.97)	16.25 (6.23,41.67)	101.97 (38.65, 230.23)	<0.001 ***
ASPECTS	8.00 (6.00, 9.00)	8.00 (6.00, 9.00)	7.00 (5.00, 9.00)	0.197
4D CTA-CS scores	3.00 (2.00, 4.00)	3.00 (3.00, 4.00)	2.00 (1.00, 3.00)	<0.001 ***
CBS	6.00 (3.00, 9.00)	6.00 (4.00, 9.00)	6.00 (2.00, 9.00)	0.135
Thrombus location				0.023 *
ICA	54 (29.67)	33 (33.33)	21 (25.30)	
Segment M1	68 (37.36)	34 (34.34)	34 (40.96)	
Segment M2	39 (21.43)	25 (25.25)	14 (16.87)	
A1	8 (4.40)	5 (5.05)	3 (3.61)	
Tandem occlusion	13 (7.14)	2 (2.02)	11 (13.25)	
Laboratory parameters, median (IQR)				
Glucose, mmol/L	7.50 (6.20, 9.63)	7.50 (6.30, 9.30)	7.50 (6.10, 9.80)	0.725
Creatinine, umol/L	76.00 (64.75, 88.25)	74.00 (64.00, 86.00)	78.00 (65.00, 100.00)	0.312
Urea, mmol/L	5.64 (4.23, 7.24)	5.49 (4.26, 6.72)	5.98 (4.07,7.86)	0.171
Uric acid, mmol/L	324.00 (252.00, 409.25)	310.00 (246.00,381.00)	339.00 (263.00, 426.00)	0.101
Sodium, mmol/L	140.10 (138.18, 142.00)	140.50 (138.50, 142.40)	139.70 (137.90, 141.70)	0.214
Potassium, mmol/L	4.00 (3.70,4.30)	4.00 (3.70,4.20)	4.00 (3.70, 4.40)	0.400
D-dimer	282.50 (140.50, 662.00)	198.00 (81.00, 373.00)	572.00 (248.00, 1414.00)	<0.001 ***
Fibrinogen, g/L	3.05 (2.65, 3.64)	3.06 (2.66, 3.52)	3.01 (2.65, 3.72)	0.484
INR	0.98 (0.92, 1.05)	0.97 (0.91, 1.05)	0.98 (0.94, 1.05)	0.093
RBC	4.40 (3.92, 4.85)	4.44 (3.96, 4.88)	4.39 (3.85, 4.83)	0.383
WBC	8.04 (6.35,10.21)	7.82 (6.34, 8.99)	8.28 (6.36, 10.91)	0.288
BNP, median (IQR)	212.31 (59.87, 477.26)	91.65 (26.91, 203.99)	353.28 (203.99, 689.63)	<0.001 ***
Time, min; median (IQR)				
Onset to imaging	251.00 (138.25, 449.75)	272.50 (144.00, 564.00)	205.00 (130.00, 346.00)	0.037 *
Imaging to puncture	76.50 (56.75, 106.00)	79.00 (56.00, 107.00)	74.00 (57.00, 106.00)	0.632
Puncture to recanalization	81.00 (52.00, 135.75)	73.00 (50.00, 125.00)	87.00 (54.00, 147.00)	0.219
Recanalization, n (%)	155 (85.16)	91 (91.92)	64 (77.11)	0.006 **
Classification, n (%)				<0.001 ***
CE stroke	77 (42.31)	29 (29.29)	48 (57.83)	
LAA stroke	105 (57.69)	70 (70.71)	35 (42.17)	

mRS: modified Rankin Scale; IQR: interquartile range; NIHSS: National Institutes of Health Stroke Scale; IV-Tpa: intravenous tissue type plasminogen activator; SBP: systolic pressure; DBP: diastolic pressure; AF: atrial fibrillation; CHD: coronary heart disease; IC: ischemic core; IP: ischemic penumbra; MMR: mismatch ratio; FIV: final infarct volume; ASPECTS: Alberta Stroke Program Early CT Score; 4D CTA-CS: the modified collateral circulation scoring system on 4D CTA; CBS: clot burden score; ICA: internal carotid artery; M1: M1 segment middle cerebral artery; M2: M2 segment middle cerebral artery; A1: A1 segment anterior cerebral artery; INR: activated partial thromboplastin time; RBC: red blood cell; WBC: white blood cell; BNP: B-type brain natriuretic peptide; CE: cardioembolic; LAA: large-artery atherosclerosis. * *p* < 0.05, ** *p* < 0.01 and *** *p* < 0.001.

**Table 2 brainsci-13-00539-t002:** Baseline characteristics of patients with CE stroke and LAA stroke.

Characteristics	LAA Stroke (*n* = 105)	CE Stroke (*n* = 77)	*p* Value
Age, y; median (IQR)	76.00 (63.75, 83.25)	81.00 (71.50, 86.00)	<0.001 ***
Female, n (%)	38 (36.19)	41 (53.25)	0.032 *
NIHSS, median (IQR)	12.50 (7.00, 17.00)	15.00 (10.00, 20.00)	<0.001 ***
IV-Tpa, n (%)	21 (20.00)	22 (28.57)	0.243
SBP, median (IQR)	145.00 (132.75,159.25)	147.00 (136.00,162.50)	0.031 *
DBP, median (IQR)	80.00 (71.00, 88.00)	80.00 (71.50, 93.50)	0.320
Risk factors, n (%)			
AF	6 (5.71)	69 (89.61)	<0.001 ***
Hypertension	78 (74.29)	62 (80.52)	0.419
Diabetes mellitus	51 (48.57)	22 (28.60)	0.010 *
Hyperlipidemia	49 (46.67)	34 (44.20)	0.853
Smoking	38 (36.19)	15 (19.48)	0.022 *
Previous stroke	47 (44.76)	35 (45.50)	1.000
CHD	40 (38.10)	39 (50.60)	0.124
Imaging examinations, median (IQR)/n (%)			
IC volume, mL	22.55 (8.99, 57.65)	39.52 (14.56, 80.77)	<0.001 ***
IP volume, mL	82.94 (42.92, 128.05)	92.73 (53.66, 158.19)	0.014 *
MMR	3.04 (1.76, 6.25)	2.29 (1.41, 4.55)	0.013 *
FIV, mL	39.44 (11.05, 101.97)	39.44 (11.05,101.97)	0.004 **
ASPECTS	8.00 (6.00, 9.00)	8.00 (5.00, 9.00)	0.582
CBS	6.00 (3.00, 9.00)	6.00 (2.50, 9.00)	0.127
4D CTA-CS scores	3.00 (2.00, 4.00)	2.00 (1.00, 3.00)	<0.001 ***
Thrombus location			
ICA	32 (30.48)	22 (28.57)	0.074
Segment M1	35 (33.33)	33 (42.86)	
Segment M2	20 (19.05)	19 (24.68)	
A1	7 (6.77)	1 (1.30)	
Tandem occlusion	11 (10.48)	2 (2.60)	
Laboratory parameters, median (IQR)			
Glucose, mmol/L	7.50 (6.20, 9.63)	7.20 (6.15, 8.25)	0.026 *
Creatinine, umol/L	76.00 (64.75, 88.25)	76.00 (66.00, 88.00)	0.878
Urea, mmol/L	5.64 (4.23, 7.24)	5.64 (4.28, 7.18)	0.820
Uric acid, mmol/L	324.00 (252.00, 409.25)	324.00 (252.00, 409.25)	0.061
Sodium, mmol/L	140.10 (138.18,142.00)	140.00 (138.30, 141.15)	0.351
Potassium, mmol/L	4.00 (3.70, 4.30)	4.00 (3.70, 4.30)	0.551
D-dimer	282.50 (140.50, 662.00)	456.00 (206.50, 914.00)	0.002 **
Fibrinogen, g/L	3.05 (2.65, 3.64)	3.01 (2.63, 3.54)	0.415
INR	0.98 (0.92, 1.05)	0.99 (0.95, 1.10)	0.007 **
RBC	4.40 (3.92,4.85)	4.30 (3.92, 4.78)	0.287
WBC	8.04 (6.35, 10.21)	7.62 (5.91, 9.64)	0.029 *
BNP	212.31 (59.87, 477.26)	466.38 (252.33, 738.68)	<0.001 ***
Time, min; median (IQR)			
Onset to imaging	251.00 (138.25, 449.75)	191.00 (117.50, 294.00)	<0.001 ***
Imaging to puncture	76.50 (56.75, 106.00)	73.00 (55.50, 105.00)	0.386
Puncture to recanalization	81.00 (52.00, 135.75)	67.00 (47.50, 100.50)	0.004 **
Recanalization, n (%)	91 (86.67)	64 (83.12)	0.649
mRS score, median (IQR)	2.00 (0.00, 4.00)	4.00 (2.00, 5.00)	<0.001 ***
Clinical outcome, n (%)			
Good outcome (mRS ≤ 2)	70 (66.70)	29 (37.70)	<0.001 ***
Poor outcome (mRS > 2)	35 (33.30)	48 (62.30)	

LAA: large-artery atherosclerosis; CE: cardioembolic; IQR: interquartile range; NIHSS: National Institutes of Health Stroke Scale; IV-Tpa: intravenous tissue-type plasminogen activator; SBP: systolic pressure; DBP: diastolic pressure; AF: atrial fibrillation; CHD: coronary heart disease; IC: ischemic core; IP: ischemic penumbra; MMR: mismatch ratio; FIV: final infarct volume; ASPECTS: Alberta Stroke Program Early CT Score; 4D CTA-CS: the modified collateral circulation scoring system on 4D CTA; CBS: clot burden score; ICA: internal carotid artery; M1: M1 segment middle cerebral artery; M2: M2 segment middle cerebral artery; A1: A1 segment anterior cerebral artery; INR: activated partial thromboplastin time; RBC: red blood cell; WBC: white blood cell; BNP: B-type brain natriuretic peptide; mRS: modified Rankin Scale. * *p* < 0.05, ** *p* < 0.01 and *** *p* < 0.001.

**Table 3 brainsci-13-00539-t003:** Multivariate analysis model.

Variables	Unadjusted OR	*p* Value	Adjusted OR	*p* Value	Adjusted OR	*p* Value
	(95% CI)		(95% CI) *		(95% CI) ^†^	
All patients (*n* = 182)						
Onset to imaging	1.001 (1.000–1.001)	0.214	1.001(1.000–1.002)	0.085	1.001 (1.000–1.002)	0.030
NIHSS	1.043 (0.966–1.126)	0.287	1.021 (0.945–1.103)	0.603	1.002 (0.920–1.091)	0.964
Classification	0.804 (0.312–2.077)	0.653	0.502 (0.182–1.390)	0.185	0.261 (0.038–1.775)	0.170
BNP	1.003 (1.001–1.005)	<0.001	1.003 (1.001–1.005)	0.001	1.004 (1.002–1.006)	0.001
IC volume	0.999 (0.990–1.008)	0.841	0.999 (0.989–1.009)	0.843	0.999 (0.988–1.010)	0.874
MMR	0.965 (0.891–1.045)	0.377	0.948 (0.872–1.031)	0.214	0.932 (0.853–1.019)	0.120
D-Dimer	1.000 (1.000–1.000)	0.885	1.000 (1.000–1.000)	0.883	1.000 (1.000–1.000)	0.608
4D CTA-CS	0.323 (0.202–0.518)	<0.001	0.281 (0.170–0.466)	<0.001	0.244 (0.138–0.432)	<0.001
Recanalization	0.366 (0.116–1.160)	0.088	0.401 (0.122–1.315)	0.132	0.364 (0.094–1.407)	0.143
CE stroke patients (*n* = 77)						
NIHSS	1.066 (0.959–1.185)	0.234	1.069 (0.948–1.206)	0.274	1.078 (0.925–1.258)	0.337
4D CTA-CS	0.503 (0.280–0.904)	0.022	0.513 (0.280–0.939)	0.030	0.395 (0.179–0.873)	0.022
IC volume	1.008 (0.992–1.024)	0.345	1.008 (0.992–1.025)	0.336	1.013 (0.991–1.035)	0.249
MMR	0.969 (0.841–1.117)	0.667	0.959 (0.828–1.111)	0.576	0.902 (0.748–1.088)	0.280
Glucose	1.187 (0.878–1.603)	0.265	1.161 (0.854–1.577)	0.341	1.124 (0.718–1.761)	0.608
D-Dimer	1.000 (1.000–1.000)	0.212	1.000 (1.000–1.000)	0.154	1.000 (0.999–0.999)	0.027
BNP	1.004 (1.001–1.007)	0.006	1.004 (1.001–1.008)	0.005	1.007 (1.003–1.012)	0.003
LAA stroke (*n* = 105)						
NIHSS	1.019 (0.902–1.150)	0.766	1.018 (0.899–1.152)	0.781	1.026 (0.870–1.198)	0.746
Recanalization	0.268 (0.045–1.579)	0.268	0.529 (0.075–3.753)	0.296	0.168 (0.014–2.058)	0.163
IV-Tpa	6.981 (1.131–43.061)	0.036	5.662 (0.822–39.008)	0.146	12.894 (0.818–203.274)	0.069
4D CTA-CS	0.222 (0.103–0.476)	<0.001	0.209 (0.090–0.488)	<0.001	0.122 (0.035–0.425)	0.001
MMR	0.945 (0.839–1.064)	0.351	0.947 (0.837–1.071)	0.283	0.923 (0.772–1.104)	0.382
D-Dimer	1.000 (1.000–1.000)	0.96	1.000 (1.000–1.000)	0.765	1.000 (1.000–1.000)	0.435
BNP	1.003 (1.001–1.006)	0.008	1.003 (0.999–1.006)	0.13	1.002 (0.997–1.006)	0.421

OR: odds ratio; CI: confidence interval; NIHSS: National Institutes of Health Stroke Scale; BNP: B-type brain natriuretic peptide; IC: ischemic core; MMR: mismatch ratio; 4D CTA-CS: the modified collateral circulation scoring system on 4D CTA; CE: cardioembolic; LAA: large-artery atherosclerosis; IV-Tpa: intravenous tissue-type plasminogen activator. * Adjusted for age, gender; † adjusted for age, gender, and risk factors.

**Table 4 brainsci-13-00539-t004:** The predictive value of 4D CTA-CS and BNP on clinical outcome in AIS and CE and LAA subtypes.

Predictors	BNP	4D CTA-CS	Combination of BNP and 4D CTA-CS
All			
AUC (95% CI)	0.792 (0.726–0.849)	0.818 (0.754–0.871)	0.870 (0.812–0.915)
*p* value	<0.001	<0.001	<0.001
Sensitivity, Specificity (%)	(79.52, 68.69)	(69.88, 91.92)	(79.52, 85.86)
PPV, NPV (%)	(68.04, 80.00)	(87.88, 78.45)	(82.50, 83.33)
DeLong test *p* value compared with BNP	-	0.556	0.012 *
DeLong test *p* value compared with 4D CTA-CS	0.556	-	0.003 *
CE			
AUC (95% CI)	0.751 (0.639–0.842)	0.799 (0.692–0.882)	0.863 (0.765–0.931)
*p* value	<0.001	<0.001	<0.001
Sensitivity, Specificity (%)	(62.50, 79.31)	(75.00, 82.76)	(83.33, 82.76)
PPV, NPV (%)	(83.33, 56.10)	(87.80, 66.67)	(88.89, 75.00)
DeLong test *p* value compared with BNP	-	0.507	0.026 *
DeLong test *p* value compared with 4D CTA-CS	0.507	-	0.031 *
LAA			
AUC (95% CI)	0.792 (0.702–0.865)	0.793 (0.703–0.866)	0.837 (0.752–0.902)
*p* value	<0.001	<0.001	<0.001
Sensitivity, Specificity (%)	(82.86, 70.00)	(62.86, 95.71)	(74.29, 87.14)
PPV, NPV (%)	(58.00, 89.09)	(88.00, 83.75)	(74.29, 87.14)
DeLong test *p* value compared with BNP	-	0.986	0.421
DeLong test *p* value compared with 4D CTA-CS	0.986	-	0.020 *

AUC: area under curve; CI: confidence interval; PPV: positive predictive value; NPV: negative predictive value; 4D CTA-CS: the modified collateral circulation scoring system on 4D CTA; BNP: B-type brain natriuretic peptide; CE: cardioembolic; LAA: large-artery atherosclerosis. * *p* < 0.05.

## Data Availability

All the data are available upon request from the corresponding author.

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
