# Peer review of "Collateral Circulation and BNP in Predicting Outcome of Acute Ischemic Stroke Patients with Atherosclerotic versus Cardioembolic Cerebral Large-Vessel Occlusion Who Underwent Endovascular Treatment"

_brainsci, 2023, doi:10.3390/brainsci13040539_

Round 1

Reviewer 1 Report

Comments and Suggestions for Authors

This is a well written paper on the clinical value of BNP and collaterals in cerebral circulation evaluated with angio-CT in patients admitted with acute ischemic stroke and clinical outcomes.  Insufficient or absent  flow through anterior and posterior communicating arteries facilitates stroke progression as well as exaggerates symptoms of cerebral ischemia. Introduction is clearly presented. Conclusions could be written this way only after the adjustment to multiple variables as study groups differ substantially.

 I have few comments that I would like the Authors to address. 

1. I think that the Authors should insert Table with study groups characteristics, including Cardio-embolic group and Large Artery Disease group, with p-values for comparisons. 

2. Abstract and Table 1. Please provide information on poor vs good outcomes score. I would suggest to specify directly: mRS below and equal or higher to 2 points. The same for multivariate analysis 

3. There is little information on endovascular treatment, whereas it is one of most important issues. The Authors should describe in detail what procedures were performed, was it thrombolytic or mechanical therapy. For thrombolytic treatment, I am concerned about therapy window; the Authors stated below 12 hours since the symptoms onset, but this is too long according to guidelines. Please explain.

4. The regression analysis should be adjusted for confounders, e.g. age, gender, risk factors ...

5. Discussion. I do not agree that patients with LAA and AIS have well developed collaterals. Please address this issue. Asymptomatic patients with LAA have better cerebral circulation compared to AIS patients. The Authors might find useful paper Postep Kardiol Inter 2015; 11, 4 (42): 312–317
DOI: 10.5114/pwki.2015.55602. 

6. There are many errors and typos in English, please check carefully.

Author Response

We would like to express our sincere thanks to the reviewers for the constructive and positive comments.

Replies to Reviewer 1

Comment 1: I think that the Authors should insert Table with study groups characteristics, including Cardio-embolic group and Large Artery Disease group, with p-values for comparisons.

Reply 1: Thank you very much for the suggestion. We added the new table (Table 2) with study group characteristics and p-value for comparisons in the manuscript.

Comment 2:Abstract and Table 1. Please provide information on poor vs good outcomes score. I would suggest to specify directly: mRS below and equal or higher to 2 points. The same for multivariate analysis.

Reply2: Thank you very much for your professional advice. We have made changes to the relevant part, including Abstract, table 1 and multivariate analysis.

Comment 3: There is little information on endovascular treatment, whereas it is one of most important issues. The Authors should describe in detail what procedures were performed, was it thrombolytic or mechanical therapy. For thrombolytic treatment, I am concerned about therapy window; the Authors stated below 12 hours since the symptoms onset, but this is too long according to guidelines. Please explain.

Reply 3: (1) According to your suggestion, we have added an introduction of surgical treatment in the Methods section (2.2.3 Related factors for EVTs). (2)All patients enrolled in this study were AIS patients undergoing endovascular therapy. In recent years, based on the DAWN study and DEFUSE3 study, mechanical thrombectomy's treatment time window has been extended to 24 hours from symptom onset, and it is safe and effective for eligible patients to undergo mechanical thrombectomy. The guidelines for endovascular treatment of acute ischemic stroke were revised in 2018. Therefore, the time window for inclusion in this study is 24 hours.

References:

[1]Nogueira RG, Jadhav AP, Haussen DC, et al. Thrombectomy 6 to 24 Hours after Stroke with a Mismatch between Deficit and Infarct. N Engl J Med. 2018;378(1):11-21. doi:10.1056/NEJMoa1706442

[2]Albers GW, Marks MP, Kemp S, et al. Thrombectomy for Stroke at 6 to 16 Hours with Selection by Perfusion Imaging. N Engl J Med. 2018;378(8):708-718. doi:10.1056/NEJMoa1713973

[3]Lee KJ, Kim BJ, Han MK, et al. Predictive Value of Pulse Pressure in Acute Ischemic Stroke for Future Major Vascular Events. Stroke. 2018;49(1):46-53. doi:10.1161/STROKEAHA.117.019582

[4]Huo X, Ma G, Tong X, et al. Trial of Endovascular Therapy for Acute Ischemic Stroke with Large Infarct [published online ahead of print, 2023 Feb 10]. N Engl J Med. 2023;10.1056/NEJMoa2213379. doi:10.1056/NEJMoa2213379

Comment 4: The regression analysis should be adjusted for confounders, e.g. age, gender, risk factors ...

Reply 4: Thank you for your professional statistical advice. We have made modifications to the table 3 and the Results section (3.2), adding adjusted odds ratios.

Comment 5: Discussion. I do not agree that patients with LAA and AIS have well developed collaterals. Please address this issue. Asymptomatic patients with LAA have better cerebral circulation compared to AIS patients. The Authors might find useful paper Postep Kardiol Inter 2015; 11, 4 (42): 312317

DOI: 10.5114/pwki.2015.55602.

Reply 5: Thank you for your advice. I guess your question is about comparing collateral circulation in patients with CE to those with LAA, and you doubted that patients with LAA had better collateral circulation. I agree with you that asymptomatic patients with LAA have better cerebral circulation compared to AIS patients. But in this study, all enrolled patients were symptomatic AIS patients within 24 hours. We found that the collateral circulation in the LAA group was better than that of the CE group patients. We did not compare patients with symptomatic and asymptomatic stroke.

In addition, what differs from the reference you provided is: (1) According to the compensatory pathways of blood flow, cerebral collateral circulation is mainly divided into three levels: the first level is the primary collateral circulation compensation, which is the Willis circle and is the most important collateral circulation pathway in the intracranial region; the second level is the secondary collateral circulation compensation, mainly including the ophthalmic artery and the first layer of the meninges collateral circulation; the third level refers to the development of neovascularization. When the secondary compensatory pathway still cannot meet the demand for blood supply, neovascularization become the final collateral circulation pathway. It is not enough to only evaluate the Willis circle for AIS patients as in this reference. (2) In our study, 4D CTA was used to evaluate the entire cerebral collateral circulation (as shown in the figure and literature below), including collateral circulation at all three levels. The reference provided only used ultrasound to evaluate the Willis circle. 4D CTA provides a more comprehensive and accurate evaluation of collateral circulation (please find our previous study in the below).

However, we are very grateful for the literature you provided, and we agree that collateral compensation differs between the recent and long-term patients. We have added this point to the limitations section.

Reference: Cao R, Qi P, Liu Y, Ma X, Shen Z, Chen J. Improving Prognostic Evaluation by 4D CTA for Endovascular Treatment in Acute Ischemic Stroke Patients: A Preliminary Study. J Stroke Cerebrovasc Dis. 2019 Jul;28(7):1971-1978. doi: 10.1016/j.jstrokecerebrovasdis.2019.03.038. Epub 2019 Apr 10. PMID: 30981581.

Comment 6:  There are many errors and typos in English, please check carefully.

Reply 6: Thank you for your suggestion. The spelling and syntax errors have been checked and corrected in the revised version.

Reviewer 2 Report

Comments and Suggestions for Authors

The authors present the results of a retrospective, single-center, clinical study analyzing the outcomes of 182 acute ischemic stroke patients (77 with cardioembolic stroke and 105 with large atherothrombotic infarction) who underwent endovascular treatment. The results demonstrated that collateral status and serum BNP could be used as independent predictors of outcome in ischemic stroke. Patients with good collateral status in both the cardioembolic and thrombotic groups had a higher proportion of good outcomes than patients with poor collateral status. BNP in the cardioembolic group, but not in the atherothrombotic group, had a predictive value on clinical outcome This report is potentially interesting, but the manuscript can be improved according to the following suggestions:  

  1. The authors should point out in the Introduction that clinical features observed at stroke onset can help to distinguish cardioembolic from atherothrombotic infarctions (Eur J Neurol 1999; 6: 677-683). The inclusion and comment on this reference is recommended) 
  2. In the Results (ROC curve), it should be interesting to add the positive predictive value and the negative predictive value. 
  3. It would be interesting to emphasize in the text that a future line of research on the discussed topic would be to study the relationship and relevance of the impact of collateral circulation and BNP as predictors of outcome in different ischemic stroke subtypes. This recommendation is due to the fact that the pathophysiology, prognosis and clinic of ischemic small vessel strokes are different from the rest of cerebral infarcts (see and add this reference: Int J Mol Sci 2022; 23, 1497).
  4. Please check reference #3 

Author Response

We would like to express our sincere thanks to the reviewers for the constructive and positive comments.

Replies to Reviewer 2

Comment 1: The authors should point out in the Introduction that clinical features observed at stroke onset can help to distinguish cardioembolic from atherothrombotic infarctions (Eur J Neurol 1999; 6: 677-683). The inclusion and comment on this reference is recommended) 

Reply 1: Thank you very much for your advice, which is very helpful in improving the content of our article. We have made revisions to the introduction section (Page1-2) and added this reference.

Comment 2:In the Results (ROC curve), it should be interesting to add the positive predictive value and the negative predictive value. 

Reply2: Thank you for your professional statistical advice. We have made modifications to the table 4, adding positive predictive value and the negative predictive value.

Comment 3:It would be interesting to emphasize in the text that a future line of research on the discussed topic would be to study the relationship and relevance of the impact of collateral circulation and BNP as predictors of outcome in different ischemic stroke subtypes. This recommendation is due to the fact that the pathophysiology, prognosis and clinic of ischemic small vessel strokes are different from the rest of cerebral infarcts (see and add this reference: Int J Mol Sci 2022; 23, 1497).

Reply 3: Thank you very much for your suggestion. We have made revisions to the Discussion section (Page 12) and added this reference.

Comment 4:Please check reference #3 

Reply 4: Thank you very much for your meticulous review. We have replaced the reference#3.

Round 2

Reviewer 1 Report

Comments and Suggestions for Authors

The manuscript was significantly improved. I appreciate adding Table for comparisons between group with LAA and CE strokes. The Authors answered the majority of comments in satisfactory way . I believe that paper was significantly improved. Still, I think that methods must be corrected. Expanding therapeutic window for mechanical and pharmacological treatment in acute stroke patients is accetable, but not in all patients. Specific additional imaging and clinical conditions must be met. I would suggest to improve methods and state that these conditions were met. As, modified guideliness were published, please see below. 

In 2015, based on the evidence of five clinical trials, the American Heart Association/American Stroke Association (AHA/ASA) included mechanical thrombectomy (MT) within 6 h of symptom onset as a therapy with level IA evidence, for patients with AIS due to a LVO [Powers WJ, et al. 2015 American heart association/American stroke association focused update of the 2013 guidelines for the early management of patients with acute ischemic stroke regarding endovascular treatment. Stroke. 2015;46:3020-35].  After several trials showed that this therapy could be used for selected patients in even longer time windows, the 2019 update of these guidelines adopted an extended therapeutic window [Powers WJ, et al. Guidelines for the early management of patients with acute ischemic stroke:2019 update to the 2018 guidelines for the early management of acute ischemic stroke:a guideline for healthcare professionals from the American heart association/American stroke. Stroke. 2019;50:3331-2.]. MT requires a selection of patients based on clinical and imaging criteria, including the need for vessel imaging with computed tomography angiography (CTA) or magnetic resonance angiography (MRA) in the 6-h window and an evaluation of mismatch between the stroke core and the area of tissue at risk using magnetic resonance imaging (MRI), computed tomography perfusion (CTP) and/or perfusion-weighted imaging MRI (Magnetic resonance perfusion [MRP] Yuh WT, et al. Revisiting current golden rules in managing acute ischemic stroke:evaluation of new strategies to further improve treatment selection and outcome. Am J Roentgenol. 2017;208:32-41. Kamalian S, Lev MH. Stroke imaging. Radiol Clin North Am. 2019;57:717-32.

In conclusion, 24 h therapeutic window is not accepted in AIS patients. 

In addition, I believe that ultrasonographic data concerning collaterals in cerebral circulation are important , as ultrasonography adds to CTA because it shows hemodynamics. therfore, a short paragraph on ultrasonography in acute ischemic stroke patients vs asymptomatic patients would be valuable

Author Response

Thank you so much for approving the changes made to the manuscript and for sharing your professional expertise with us. We have gained valuable insights from your contributions. We have made the following modifications: (1) We have made corrections to the methods section by incorporating the literature you provided and by adding the inclusion criteria ( Methods section 2.1). (2) We have provided supplementary explanations for ultrasonography, summarized its significance in a paragraph, and cited references in the Discussion section.